Sexual reproduction in the Caribbean coral genus Isophyllia (Scleractinia: Mussidae)

Soto Derek derek.soto@upr.edu
Weil Ernesto
Department of Marine Science, Universidad de Puerto Rico, Recinto de Mayagüez , Mayagüez, Puerto Rico , United States
Pochon Xavier
Electronic publication date: 2016 Nov 10
Publication date: 2016
Volume: 4
Electronic Location ID: e2665
Received 2015 Oct 5; Accepted 2016 Oct 7
Copyright: © 2016 Soto and Weil
Copyright year: 2016
Copyright holder: Soto and Weil
License: This is an open access article distributed under the terms of the Creative Commons Attribution License, which permits unrestricted use, distribution, reproduction and adaptation in any medium and for any purpose provided that it is properly attributed. For attribution, the original author(s), title, publication source (PeerJ) and either DOI or URL of the article must be cited.
License URL: https://creativecommons.org/licenses/by/4.0/

Keywords: Caribbean, Mussidae, Coral reproduction, Hermaphroditic, Brooder

Funding: Deep CRES-NOAA Coastal Ocean Program award # NA09NOS4260223 Sea Grant College Program grant # R-101-1-98 Funding was provided by Deep CRES-NOAA Coastal Ocean Program award # NA09NOS4260223 to the University of Puerto Rico and a Sea Grant College Program grant (# R-101-1-98) to E. Weil provided partial funding, equipment and/or logistics. The funders had no role in study design, data collection and analysis, decision to publish, or preparation of the manuscript.

==============================
The sexual pattern, reproductive mode, and timing of reproduction of Isophyllia sinuosa and Isophyllia rigida, two Caribbean Mussids, were assessed by histological analysis of specimens collected monthly during 2000–2001. Both species are simultaneous hermaphroditic brooders characterized by a single annual gametogenetic cycle. Spermatocytes and oocytes of different stages were found to develop within the same mesentery indicating sequential maturation for extended planulation. Oogenesis took place during May through April in I. sinuosa and from August through June in I. rigida. Oocytes began development 7–8 months prior to spermaries but both sexes matured simultaneously. Zooxanthellate planulae were observed in I. sinuosa during April and in I. rigida from June through September. Higher polyp and mesenterial fecundity were found in I. rigida compared to I. sinuosa. Larger oocyte sizes were found in I. sinuosa than in I. rigida, however larger planula sizes were found in I. rigida. Hermaphroditism is the exclusive sexual pattern within the Mussidae while brooding has been documented within the related genera Mussa, Scolymia and Mycetophyllia. This study represents the first description of the sexual characteristics of I. rigida and provides an updated description of I. sinuosa.

Introduction

Reproduction in corals consists of a sequence of events which include: gametogenesis, spawning (broadcasters), fertilization, embryogenesis, planulation (brooders), dispersal, settlement and recruitment (Harrison & Wallace, 1990). The success of the reproductive effort is determined largely by the timing, duration, frequency and intensity of the aforementioned events (Babcock et al., 1986). In corals, sexual pattern, mode of reproduction, fertilization, larval dispersal, recruitment and survivorship are key components in determining evolutionary fitness (Szmant, 1986; Edmunds, 2005; Vermeij, 2006; Weil, Croquer & Urreiztieta, 2009; Pinzon & Weil, 2011) which is defined as the product of sexual output (fecundity) and survivorship (Metz, Nisbet & Geritz, 1992). Consequently, the ability of coral species to adapt to modern-day environmental pressures depends greatly on the ability of species to reproduce effectively.

The reproductive characteristics of some scleractinian groups have been more thoroughly studied than others; however, little is known about the reproductive patterns of many Caribbean coral species and some of the available information is conflictive or incomplete (Fadlallah, 1983; Harrison & Wallace, 1990; Harrison, 2011; Weil & Vargas, 2010; Pinzon & Weil, 2011). Of the approximately 60 Caribbean zooxanthellate coral species reported, thorough descriptions of their reproductive characteristics and cycles are available for 19 species; many other studies available provide partial or conflicting results (Weil, 2003; Weil & Vargas, 2010; Harrison, 2011). Reproductive studies of the sexual patterns of I. sinuosa were among the first studies of such nature performed in the Caribbean (Duerden, 1902). These were limited to histological observations of oocytes in a few colonies of I. sinuosa (Figs. 1A and 1B), therefore, the species is classified as gonochoric. This characterization contrasts with the reproductive mode of other studied Mussids which are classified as hermaphroditic. Currently, there is no information available on the reproductive biology of I. rigida (Figs. 1C and 1D).

Figure 1 Plate showing study corals.

(A, B) Isophyllastrea rigida (C, D) Isophyllia sinuosa. Photos by Ernesto Weil.

This study characterizes the reproductive biology of I. rigida and I. sinuosa in terms of sexual pattern, mode of development, gametogenic cycles, and fecundity. These fundamental aspects of the physiology of this taxa are understudied. Knowledge of the reproductive biology and ecology of coral species is important for the interpretation of their population and ecological dynamics, their patterns/potential for dispersal, and their local and geographical distribution. The threats currently faced by coral reefs and the ongoing global effort to understand why corals are dying highlight the need to expand our understanding of basic coral physiology.

Materials and Methods

Sampling for this study was carried out at La Parguera Natural Reserve, off the southwest coast of Puerto Rico (Fig. 2). This complex reef environment is among the many regions experiencing deterioration by anthropogenic and environmental climate influences at local and global scales. Coral reefs in La Parguera are important local economic drivers, supporting artisanal and recreational fishing, tourism, recreational activities and also protect coastal settlements, seagrass communities and other wetland habitats from the effects of hurricanes and coastal erosion (Ballantine et al., 2008).

Figure 2 Map of La Parguera, Puerto Rico with study sites.

Image made with QGIS using NOAA’s National Centers for Enviromental Information (NCEI) Multibeam Bathymetric Surveys Dataset.

At least five unique sample cores were collected monthly for 14 months between March 2000 and May 2001 (Fig. 3A). A total of 89 samples of each species were collected. Colonies were selected by searching in a zig-zag pattern over the distributional range of both species (5–18 m). Samples were collected from San Cristobal reef (17°55′24.88″N, 67°6′14.52″W), Caracoles reef (17°57′46.02″N, 67°2′8.21″W), Media Luna reef (17°56′22.68″N, 67°2′43.26″W), Pinaculos (17°56′1.13″N, 67°0′39.75″W), Turrumote reef (17°56′13.56″N, 67°1′8.92″W), Beril (17°52′47.85″N, 66°59′1.40″W), El Palo (17°55′50.2″N, 67°05′36.9″W), Laurel (17°55′50.2″N, 67°05′36.9″W) and Enrique (17°55′50.2″N, 67°05′36.9″W) (Figs. 3B and 4).

Figure 3 Collection data for [i]I. sinuosa[i] and [i]I. rigida[i].

(A) Number of samples collected per month, (B) Number of samples collected per location.

Figure 4 Developmental stages of oocytes (O) and spemaries (S) in I. sinuosa.

(A) Stage I and II oocytes, (B) stage III oocytes, (C) stage II spermaries and stage IV oocytes, (D) stage IV oocytes and stage III spermaries, (E) stage IV oocytes and stage V spermaries, and (F) stage II planula. Scale bar measures 100 μm2.

Sample cores were placed in Zenker Formalin (Helly’s solution) for 24 h, rinsed and then decalcified in 10% HCl solution. Tissues were then cleaned and placed in plastic tissue holders. Preserved samples were sequentially dehydrated in the rotary tissue processor under 70 and 95%, ethanol, Tissue Dry, and xylene solution (Tissue Clear III). Samples were embedded into Paraplast blocks then sectioned using a rotary microtome. The 8–10 strip sections (7–10 μm) were obtained from each embedded block and placed onto glass slides. Finished tissue slides were stained utilizing a modified Heidenhain’s Aniline-Blue method (Coolidge & Howard, 1979) to examine the maturation stages of gametocytes and embryos.

Slides were examined under an Olympus BX40 compound microscope coupled to an Olympus DP26 digital microscope camera. Images were captured utilizing Olympus cellSens 1.7 imaging software. The sexual pattern, gametogenic cycle and fecundity of each species were determined by observing the gametocyte development throughout the collection year. Gamete stages were characterized according to Szmant-Froelich, Reutter & Riggs (1985). Oocyte sizes were obtained using cellSens, by taking perpendicular measurements at the cell’s widest points. Cell length and width measurements were used to calculate geometric area. Fecundity was assessed by counting oocytes per mesentery (I. sinuosa n = 120; I. rigida n = 60) and per polyp (I. sinuosa n = 10; I. rigida n = 5) on histologic cross-sections during months with the highest proportion of mature oocytes (I. sinuosa April 2001 n = 5; I. rigida May 2001 n = 5).

In April 2012, several presumed gravid colonies of each species were collected and placed in an open seawater aquarium system to observe planulation. Two colonies of each species were placed within six-gallon aerated aquariums under continuously circulating seawater and daylight synchronized lights. Specimens were placed under mesh-lined PVC pipes allowing water to freely circulate. Traps were checked daily for larvae over a 90-day period.

Statistical analyses

Results are expressed as means ± standard error. All statistical tests were performed using the RStudio 0.99.484 software platform (R Studio Team, 2015) using the stats package (R Development Core Team, 2015). Normality was assessed using the Shapiro-Wilk test performed with the R function shapiro.test. Equality of variance was tested using the F test performed with the R function var.test. Differences in fecundity were tested by means of a Wilcoxon rank sum test with continuity correction performed with the R function wilcoxon.test.

Collection permit

All coral tissue samples were collected under a General Collection Permit granted by the Puerto Rico Department of Natural Resources (DNER) to the Faculty of the Department of Marine Sciences, University of Puerto Rico Mayaguez (UPRM).

Results

I. sinuosa

Stage I oocytes are small (78.92 ± 13.15 μm2), stain pink and are characterized by sparse cytoplasm and prominent nuclei (Fig. 4A). Oocytes originate within the linings of the mesoglea in the central regions of the mesenteries. Stage II oocytes are larger than stage I cells (144.54 ± 43.19 μm2), exhibit prominent nuclei and abundant cytoplasm (Fig. 4A). Stage III oocytes are larger than stage II (264.51 ± 37.24 μm2), tend to have a round shape, stain pink or red, and are characterized by many cytoplasmic globules which produce a grainy appearance (Fig. 4B). Stage IV oocytes are larger and boxier than stage III (376.69 ± 73.20 μm2). This stage is characterized by dark staining nuclei and large globules in the cytoplasm (Figs. 4C–4E).

No stage I spermaries were found, suggesting this stage occurs briefly and/or is difficult to differentiate using the current method. Stage II spermaries form small poorly defined bundles which form in the mesenteries surrounding oocytes (Fig. 4C). Stage III spermaries form small sacs with well-defined borders (Fig. 4D) and contain bright red staining spermatids. Stage IV spermaries stain dark red and are larger than stage III. Tails visible on spermatozoa at high magnification are indicative of stage V spermaries (Fig. 4E). Spermary sizes were not measured.

Stage I planulae are approximately the same size as stage IV oocytes (404.07 μm2) and stain pink. During this stage, zooxanthellae become visible within the planulae. Stage II planulae (455.45 ± 32.84 μm2) are characterized by an outer layer composed of columnar cells which contain nematocysts and cilia (Fig. 4F). Developing mesenteries can be seen within the gastrodermis of stage III planula (501.98 ± 44.68 μm2). Stage IV planula were not observed.

The gametogenic cycle of I. sinuosa is summarized in Fig. 5. Weekly sea surface temperature measurements taken during the collecting period are included for reference (Fig. 5A). Oogenesis in I. sinuosa lasts approximately 11 months (Fig. 5B). Onset of oogenesis was determined to occur during May 2000 and during April 2001. Onset of oogenesis was determined as the month of appearance of stage I and II oocytes after the culmination of the previous gametogenic cycle. Stage II oocytes were prevalent in tissues during all months sampled except during November 2000 and January 2001. Stage III oocytes were observed in all sampled months except April 2001. Stage IV oocytes were observed between August 2000 through May 2001.

Figure 5 Gametogenic cycle of I. sinuosa.

(A) Sea surface temperature ranges in La Parguera, Puerto Rico. Adjusted values of relative proportions of colonies of I. sinuosa in each gametogenetic stage of (B) oogenesis, (C) spermatogenesis, and (D) embryogenesis from March 2000 to May 2001. O-I, O-II, O-III, O-IV represent oocyte stages 1 through 4, respectively; S-I, S-II, S-III, S-IV, S-V represent spermary stages 1 through 5, respectively; P-I, P-II, P-III, P-IV represent planulae stages 1 through 4, respectively.

Spermatogenesis takes places during four months (Fig. 5C). Onset of spermatogenesis was not determined because stage I spermaries were not identified. Stage II spermaries were observed during January through February 2001. Stage III spermaries were visible from January through March 2001. Stage IV spermaries were present in March 2001. Stage V spermaries were present in tissues in April 2001.

Stage I–III planulae were observed in histologic sections during April 2001 (Fig. 5D). The identification of planulae on tissue sections coincided with a sharp decrease in the proportion of colonies containing mature (IV) oocytes. No larvae were collected from specimens placed in aquaria for observation.

I. rigida

Stage I oocytes are very small (72.97 ± 15.75 μm2) and are characterized by sparse cytoplasm and a large nucleus. Stage II oocytes are larger than stage I cells (101.25 ± 23.09 μm2), are ovoid shaped and feature a prominent nucleus and nucleolus (Fig. 6A). A pink-staining nucleus and red nucleolus can clearly be identified in many stage III oocytes (148.77 ± 49.35 μm2) (Fig. 6B). Stage IV oocytes are large (190.40 ± 45.18 μm2), irregularly shaped and contain large vacuoles in the ooplasma which give it a grainy appearance (Figs. 6C and 6D).

Figure 6 Developmental stages of oocytes (O) and spemaries (S) in I. rigida.

(A) Stage II oocytes in the mesoglea, (B) stage III oocytes and Stage II spermaries, (C) stage III spermaries and stage IV oocytes, (D) stage IV oocytes and spermaries, (E) stage V spermaries, and (F) stage II planula. Scale bar measures 100 μm2.

Stage I spermaries were not detected in I. rigida. Stage II spermaries were observed forming adjacent to stage III eggs (Fig. 6B). Spermaries typically adopt a spherical shape and often form in series resembling a string of beads (Figs. 6B and 6C). Stage III spermaries form small oblong sacs and stain red (Fig. 6C). Stage IV spermaries are densely packed with sperm, have irregular shapes, stain dark red to brown. Stage V spermaries stain darker than stage IV (Fig. 6E) but are characterized by tails on spermatozoa under high magnification. No measurements were collected for spermaries.

Stage I planulae are approximately the same size as stage IV oocytes (approximately 324.01 ± 71.64 μm2), stain pink, and contain zooxanthellae in the epidermis. Zooxanthellae were observed within planula beginning at this stage. Stage II planulae are larger (521.27 ± 84.18 μm2) (Fig. 6F) and exhibit an epidermis consisting of columnar epithelium similar to I. sinuosa. Stage III and stage IV larvae measure 818.91 ± 82.96 μm2 and 951.78 ± 176.36 μm2 respectively, and show clear development of the mesenteries.

The gametogenic cycle of I. rigida is summarized in Fig. 7. Weekly sea surface temperature measurements taken during the collecting period are included for reference (Fig. 7A). Oogenesis in I. rigida lasts approximately 10 months (Fig. 7B). Oogenesis began during August 2000. Stage II oocytes were observed in tissues in March 2000 and August 2000 to April 2000. Stage III oocytes were observed in March 2000, May and June 2000 and from January 2001 through May 2001. Stage IV oocytes were observed in samples collected during March, May and June 2000, and February, April and May 2001.

Figure 7 Gametogenic cycle of I. rigida.

(A) Sea surface temperature ranges in La Parguera, Puerto Rico. Adjusted values of relative proportions of colonies of I. sinuosa in each gametogenetic stage of (B) oogenesis, (C) spermatogenesis, and (D) embryogenesis from March 2000 to May 2001. O-I, O-II, O-III, O-IV represent oocyte stages 1 through 4, respectively; S-I, S-II, S-III, S-IV, S-V represent spermary stages 1 through 5, respectively; P-I, P-II, P-III, P-IV represent planulae stages 1 through 4, respectively.

Spermatogenesis in I. rigida is estimated to last approximately 2–3 months (Fig. 7C). Onset of spermatogenesis was not determined because stage I spermaries were not identified. Stage II spermaries were observed in May 2000. Stage III spermaries were visible in May 2000. Stage IV spermaries were observed first in June 2000. Stage V spermaries were observed in May 2000.

Stage I planulae were observed in June 2000 indicating the onset of embryogenesis (Fig. 7D). The appearance of planulae coincided with a sharp decrease in the proportion of colonies containing mature oocytes. Stage II planulae were observed during June 2000 and May 2001. Stage III planulae were observed from June through August 2000. Stage IV planulae were observed in tissues from June throughout September 2000. No larvae were collected from specimens placed in aquaria for observation.

Fecundity

Mesenterial fecundity in I. sinuosa (11.13 ± 8.27 oocytes/mesentery) was significantly higher (Wilcoxon-rank sum test, W = 1,208, p < 2.2 × 10−16) than in I. rigida (1.70 ± 3.52 oocytes/mesentery) (Fig. 8A). Polyp fecundity in I. sinuosa (110.11 ± 96.33 oocytes/polyp) was significantly higher (Wilcoxon-rank sum test, W = 18, p = 0.018) compared to I. rigida (20.45 ± 23.91 oocytes/polyp) (Fig. 8B).

Figure 8 (A) Average mesenterial (eggs/mesentery) fecundity and (B) polyp (eggs/polyp) fecundity in I. sinuosa and I. rigida.

Whiskers represent standard error.

Oocyte size

Measurements of oocyte geometric area in I. sinuosa (range 43.94–463.79 μm2) show an increase in the size of oocytes as maturity progresses from April through March (Fig. 9A). Mean geometric area is lowest during the month of June 2000 (97.22 ± 28.85 μm2) and greatest during February 2001 (333.95 ± 74.32 μm2). The appearance of planulae in histological sections during the month of April 2001 (459.07 ± 45.83 μm2) (range: 404.07–548.49 μm2) coincides with a sharp decrease in mean geometric area of oocytes compared to the previous month (285.68 ± 96.46 vs. 143.28 ± 84.07 μm2). Measurements of oocyte geometric area in I. rigida (range 43.31–307.35 μm2) also show a trend of increasing oocyte size as maturity progresses from August through June (Fig. 9B). Mean geometric area is lowest during the month of September 2000 (68.35 ± 17.04 μm2) and greatest during June 2000 (210.54 ± 42.90 μm2). Mean planulae area was greatest during the month of July 2000 (909.48 ± 250.56 μm2) and ranged from 241.66–1,183.96 μm2. Mean oocyte geometric area was greater in I. sinuosa than in I. rigida (Wilcoxon-rank sum test, W = 43,911, p < 2.13 × 10−13); however, mean planulae geometric area was significantly higher in I. rigida compared to I. sinuosa (Wilcoxon-rank sum test, W = 186, p = 0.008).

Figure 9 Monthly geometric mean oocyte and planulae areas in (A) I. sinuosa and (B) I. rigida.

Discussion

Microscopic observations indicate that both I. sinuosa and I. rigida are simultaneous hermaphrodites (gametes of both sexes are present in a single individual at the same time). Gametes of both sexes are produced adjacent within the same mesentery (dygonism) in both species. Both species are brooders (bear live young) which transfer endosymbiotic zooxanthellae directly from parent to offspring. Both species are characterized by a single annual gametogenic cycle. This study represents the first description of the sexual characteristics of I. rigida and contradicts observations by Duerden (1902) which label I. sinuosa as a gonochoric species. The incorrect classification of I. sinuosa as the sole gonochoric outlier within the traditional Mussidae was a contrasting element in a group which is otherwise uniformly hermaphroditic (Duerden, 1902; Fadlallah, 1983; Richmond & Hunter, 1990). This study confirms the dominant pattern of sexual reproduction described for Mussid corals (Baird, Guest & Willis, 2009) and provides further support for conserved reproductive patterns within coral families (Harrison, 2011).

Traditional morphology-based classifications are being restructured by designating systematic affinities using molecular methods in combination with morphometric analyses. The traditional Mussidae family has recently undergone extensive restructuring by separating Indo-Pacific Mussids from their Atlantic counterparts which are more closely related to some members of the family Faviidae (Fukami et al., 2004; Fukami et al., 2008; Budd et al., 2012). The resulting ‘modern’ Mussidae (clade XXI) is composed of the genera Mussa, Isophyllia, Mycetophyllia, and Scolymia (Atlantic) under the Mussinae subfamily and Favia (Atlantic), Colpophyllia, Diploria, Pseudodiploria, Manicina and Mussismilia under the Faviinae subfamily. Under the new classification, hermaphroditism has been exclusively documented within all genera of the subfamily Mussinae: Mycetophillia (Szmant-Froelich, 1985; Morales, 2006), Scolymia (Pires, Castro & Ratto, 2002; E. Weil, 2016, unpublished data) and Mussa (Steiner, 1993) and within the subfamily Faviinae: Favia (Soong, 1991), Colpophyllia (E. Weil, 2016, unpublished data), Diploria (Weil & Vargas, 2010) Pseudodiploria (Weil & Vargas, 2010), Manicina (Johnson, 1992), Mussismilia (Pires, Castro & Ratto, 1999) (Table 1). Mode of development within the modern Mussidae is mixed; both brooding and spawning species are present. Brooding has been documented within Mycetophyllia (Morales, 2006), Scolymia (Pires, Castro & Ratto (2002); E. Weil, 2016, unpublished data), and Manicina (Johnson, 1992). Broadcast spawning occurs in Colpophyllia (E. Weil, 2016, unpublished data), Diploria (Weil & Vargas, 2010), Pseudodiploria (Weil & Vargas, 2010), and Favia (Soong, 1991). Sexual mode exhibits more plasticity than sexuality (Van Moorsel, 1983; Harrison, 1985): contrasting modes of development exist within families and even within genera (Harrison, 2011).

Table 1 Comparison of reproductive characteristics of Mussidae (Clade XXI).

Subfamily	Genus	Species	Sexual pattern	Mode of development	Source	
Mussinae	Mussa	M. angulosa	H		Steiner (1993)	
	Isophyllia	I. rigida	H	Brooding	This study	
		I. sinuosa	H	Brooding	Duerden (1902) and This study	
	Mycetophyllia	M. ferox	H	Brooding	Szmant-Froelich (1984), Szmant (1986) and Morales (2006)	
		M. aliciae	H	Brooding	Morales (2006)	
		M. lamarckiana	H	Brooding	Morales (2006)	
		M. danaana	H	Brooding	Morales (2006)	
		M. reesi				
	Scolymia (Atlantic)	S. cubensis	H	Brooding	E. Weil (2016, unpublished data)	
		S. lacera	H	Brooding	E. Weil (2016, unpublished data)	
		S. wellsi	H	Brooding	Pires, Castro & Ratto (2002)	
Faviinae	Favia (Atlantic)	F. fragrum	H	Broadcast	Duerden (1902), Fadlallah (1983), Szmant (1986), Richmond & Hunter (1990) and Soong (1991)	
	Colpophyllia	C. amaranthus	H	Broadcast	E. Weil (2016, unpublished data)	
		C. natans	H	Broadcast	Steiner (1995), Hagman, Gittings & Deslarzes (1998), Boland (1998) and E. Weil (2016, unpublished data)	
	Diploria	D.labyrinthiformis	H	Broadcast	Duerden (1902), Fadlallah (1983), Wyers, Barnes & Smith (1991) and Weil & Vargas (2010)	
	Pseudodiploria	D. clivosa	H	Broadcast	Soong (1991) and Weil & Vargas (2010)	
		D. strigosa	H	Broadcast	Szmant (1986), Richmond & Hunter (1990), Soong, 1991, Steiner (1995) and Weil & Vargas (2010)	
	Manicina	M. areolata	H	Brooding	Duerden (1902), Fadlallah (1983), Richmond & Hunter (1990) and Johnson (1992)	
	Mussismilia	M.hispida	H	Broadcast	Neves & Pires (2002) and Pires, Castro & Ratto (1999)	
		M. hartii	H	Broadcast	Pires, Castro & Ratto (1999)	
		M. brazilensis	H	Broadcast	Pires, Castro & Ratto (1999)	
Note:

H, hermaphroditic; G, gonochoric.

Szmant (1986) suggested that sexual mode is potentially a function of habitat stability, where successful recruiters would be small, rapidly maturing species, which produce many offspring over short periods but subject to high mortality rates. Thus, the sexual modality of species occupying unstable habitats would gravitate towards brooding because it increases the chances of successful recruitment by reducing gamete and larval mortality even in low population densities. Edinger & Risk (1995) on noting a correlation between brooding and eurytopy, hypothesized that brooding corals may preferentially survive in unstable habitats due to higher recruitment success. The benefits provided by the brooding modality may partially explain why, in recent decades, brooding corals have begun to dominate some Caribbean reefs following degradation by natural and anthropogenic disturbances (Hughes, 1994; Mumby, 1999; Knowlton, 2001; Irizarry-Soto & Weil, 2009).

The close proximity of oocytes and spermaries within the same mesentery (dygonism) in I. sinuosa and I. rigida suggests that it is possible that self-fertilization can occur in these species. Generally, self-fertilization is not a favored method of fertilization in corals due to possibility of inbreeding depression (Knowlton & Jackson, 1993). Selfing, however, is thought to be advantageous in sessile hermaphrodites which are ecologically distant from other mates and may have limited access to gametes of the other sex, providing a viable alternative for successful fertilization (Ayre & Miller, 2004; Darling et al., 2012; Sawada, Morita & Iwano, 2014). These corals may then switch to sexually produced larvae as population sizes increase (Ayre & Resing, 1986). Selfing has been documented in the brooding corals Seriatopora hystrix (Sherman, 2008), Favia fragum and Porites astreoides (Brazeau, Gleason & Morgan, 1998).

The duration of the gametogenic cycle is similar in I. sinuosa and I. rigida (11 and 10 months, respectively). Long oocyte generation times, differential gamete maturation, and long brood retention times in Isophyllia suggest the possibility of multiple brooding events during a single gametogenetic cycle. This strategy may increase reproductive output due to space limitations within polyps. A single annual gametogenetic cycle is the dominant pattern in most broadcasting corals such as Orbicella, Montastraea, Diploria, Porites, Acropora, Siderastrea (Szmant, 1986; (Vargas-Ángel & Thomas, 2002; Weil & Vargas, 2010) and brooding Caribbean corals like Porites and Mycetophyllia (Szmant, 1986; Soong, 1993; Vermeij et al., 2004; Morales, 2006). Multiple spawning events have been documented in Acanthastrea lordhowensis (Wilson & Harrison, 1997) and cannot be ruled out in these species.

Both species differ in the timing of oogenesis and planulogenesis events by various months which suggests that opportunities for hybridization between both species are limited. The dates of onset of oogenesis in both species (May in I. sinuosa and August in I. rigida) coincide with warm local sea surface temperatures suggesting seasonal synchronization of the gametogenic cycle. In I. sinuosa, planulae were observed in histologic sections during April 2001 which suggests that fertilization occurred during early April (most recent Full Moon: April 9). In I. rigida, planulae were observed in June 2000 which suggests a fertilization date in late May (most recent Full Moon: May 6, 2001). Various environmental factors have been shown to correlate with coral reproductive cycles and may play a role in their synchronization, including sea temperature, salinity, day length, light/dark cycles and tidal cycles (Harrison & Wallace, 1990). Van Woesik, Lacharmoise & Köksal (2006) showed experimentally that some coral spawning schedules correlate strongly with solar insolation levels prior to gamete release, however, water temperatures are highly influential in determining actual gamete maturity. van Woesik (2009) also demonstrated a positive correlation between the duration of regional wind calm periods and the coupling of mass coral spawnings. Studies with the brooding coral Pocillopora damicornis revealed that synchronization of larval production was lost under constant artificial new moon and full moon conditions, demonstrating that planulation in some species is linked to nighttime irradiance (Jokiel, Ito & Liu, 1985).

Acquisition of the endosymbiont Symbiodinium in Isophyllia occurs directly from parent to offspring (vertical transmission), a characteristic strongly linked to the brooding modality (Baird, Guest & Willis, 2009). Vertical symbiont transmission may be advantageous by providing larvae with various Symbiodinium genotypes which may improve their ability to recruit successfully and grow in different environmental conditions (Padilla-Gamiño et al., 2012). Brooded larvae are capable of motility immediately or shortly after planulation (Fadlallah, 1983), in contrast to broadcast spawned propagules which are positively buoyant and may take between 12–72 h to become motile (Baird, Guest & Willis, 2009). By avoiding the surface, brooded larvae may better avoid exposure to high levels of solar radiation which may overwhelm the photosynthetic capacities of zooxanthellae producing oxygen radicals (Tchernov et al., 2004) and cause tissue damage and mortality (Lesser et al., 1990). However, under high temperature conditions, larvae of corals with vertical symbiont transmission may suffer higher oxidative stress and tissue damage, suggesting that these corals may be more vulnerable to the effects of ocean warming (Yakovleva et al., 2009).

There is increasing evidence that sexual reproduction in corals is highly susceptible to natural and anthropogenic stressors that reduce fecundity, fertilization success, and larval survival (Harrison & Wallace, 1990; Harrison, 2011). Increases in sea surface temperatures as a consequence of global warming have produced widespread coral bleaching events and disease outbreaks with massive mortality of susceptible individuals. This worldwide decline of coral reefs underscores the need for understanding sexual reproduction in corals as the only mechanism capable of safeguarding their future. Sexual recombination is an important prerequisite for the selection of individuals which are to be able to adapt to the pressures of a changing environment. A greater understanding of the mechanisms and variables in sexual reproduction in corals, in combination with knowledge of the taxonomy and variability of the species, is essential for any coral reef management strategy (Harrison & Wallace, 1990).

Supplemental Information

Supplemental Information 1 Dataset consisting of raw oocyte, spermary and planula prevalence, oocyte sizes and fecundity obtained from histologic sections of I. sinuosa and I. rigida.

Click here for additional data file.

We would like to acknowledge I. Urreiztieta for her training and assistance with coral histology. The authors would like to acknowledge the Department of Marine Sciences, University of Puerto Rico Mayaguez (UPRM) for providing support for boat use, diving, and laboratory space. We also thank the reviewers for their helpful comments which greatly enhanced this manuscript.

Additional Information and Declarations

Competing Interests

Author Contributions

Field Study Permissions

Data Deposition

The authors declare that they have no competing interests.

Derek Soto performed the experiments, analyzed the data, contributed reagents/materials/analysis tools, wrote the paper, prepared figures and/or tables, reviewed drafts of the paper.

Ernesto Weil conceived and designed the experiments, performed the experiments, contributed reagents/materials/analysis tools, wrote the paper, reviewed drafts of the paper.

The following information was supplied relating to field study approvals (i.e., approving body and any reference numbers):

Corals were sampled under a General Collection Permit granted by the Puerto Rico Department of Natural Resources (DNER) to the Faculty of the Department of Marine Sciences UPRM.

The following information was supplied regarding data availability:

The raw data has been supplied as Supplemental Dataset Files.

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
