# Peer review of "Sexual reproduction in the Caribbean coral genus Isophyllia (Scleractinia: Mussidae)"

_PeerJ, doi:10.7717/peerj.2665_

## Round 0.1 · original submission · Major Revisions

· Academic Editor

Major Revisions

Dear Derek and Ernesto,

Both reviewers recognized the important contribution your work could make to the field of coral reef biology but highlighted that significant improvements are required before this manuscript can be considered for publication. I agree with their assessments and advise you to address each point carefully. Particular attention should be given to improving the flow, structure, and content of the introduction and discussion sections and clarifying the experimental design.

Reviewer 1 ·

Basic reporting

Generally fine, but more polish and focus is required.

Experimental design

Fine

Validity of the findings

Fine.. but don't over extend from what this study does. It is a terrific contribution in invertebrate zoology. It doesn't need a cloak of climate change and ecology to which it doesn't really speak.

Additional comments

Review of “Sexual reproduction in the Caribbean coral genus Isophyllia
(Scleractinia: Mussidae) in Puerto Rico” by Soto and Weil

This study reports on the sexual reproduction of two corals in Puerto Rico through the use of frequent sampling and histological analysis. These are classic tools being applied to long standing questions, and I am delighted to see this work being conducted and submitted for review. In the current push to understand why corals are dying and how they are affected by climate change, it is often overlooked that fundamental aspects of the biology of this taxon remain poorly studied. For this reason alone, this work should be published.

However, I would strongly encourage a revision of this work to focus on the strengths of what it can do (describe reproduction very well) and what it cannot do (address climate change effects and broader issues of coral reef ecology. For guidance, it would be valuable to look at classic histological studies of coral tissue and make sure that this work adheres to the standard that has been in place for decades. Alina Szmant, of course, has lots of examples of this kind of work, but it might also be valuable to consult work by Vandermeulen as another author that comes to mind because of good histology. Specific suggestions include:

1. Much of the early stuff in the intro does not seem relevant to this classic study. The intro could start at line 51.

2. Fecundity data needs more rigor. Please provide units for the data and provide some idea of what “mesenterial fecundity” means. What is the size of the mesenteries (length, volume?) and provide values of t and the df for each test. “Polypal” is not a real word.

3. Focus the discussion on the invertebrate biology that this paper addresses, and cut out the colloquial language (“hints at”, “cannt be discarded”, “.. capable of safeguarding their future” etc.).

4. Please add scale bars to all images. The map is not required. All the graphs need polish for a peer reviewed journal. They all look like what they are – basic graphic formats from Excel. Figs. 3 and 4 are not needed. Figs. 7 and 8 need more sophistication – look at published paper for format suggestions. Make sure the sample size is described. Table 1 seems to take a tremendous amount of space and doesn’t speak to the issues addressed in the paper.

Reviewer 2 ·

Basic reporting

Unfortunately the article “Sexual reproduction in the Caribbean coral genus Isophyllia (Scleractinia: Mussidae) in Puerto Rico” has several limitations in basic reporting. Overall the manuscript is disorganized and badly written. There are several grammatical errors, no connections between sentences or ideas, no reference for figures and some figures are in the wrong order. This makes the manuscript hard to follow and I strongly suggest the authors to find an English native speaker to review the manuscript before they can resubmit this work.

Some specific comments to improve the manuscript:
Abstract:
-Include implications of the study, why is it important?
-No need to include specifics such as what fixative or staining you used.
-Include big picture idea at the beginning and end.
-Several sentences are not clear (ex: “both sexes matured simultaneously”…spermaries were only present a few months of the year).
-Last sentence of the abstract is not well linked to the previous sentence.

Introduction
-Very disorganized, the first two paragraphs should be deleted or revised and move them after the third paragraph. Your study does not address any of the topics that you address at the beginning of the introduction.
-Start the intro with the third paragraph.
-Reference Table I in the introduction
-Include what is the significance of your study.

Materials & Methods
-What do you mean by keystone ecological component? Please revise this term….
-Line 77: remove “S” to setllements
-Line 83: replace “for” for “depending on”
-Include Fig. 3 in the text for this section
-Why did you wrap the samples in cheesecloth?
-Include the statistics that you used and if you estimated normality and equality of variance
-Is not clear to me what is your total sample size. Figures 3 and 4 are good but you also need to include a table that contains how many samples were collected at each site for each month, for each year.
-What stats did you use to compare variation in the onset, duration and culmination of the gametogenetic cycle among years and species?
-How many samples did you process to obtain your polyp/mesentery fecundity values? How many polyps per sample? How many mesenteries per polyp?
-Include spermatogenesis data in table 2.

Results
-Reference your figures in the text!
-There are several extra spaces, ex: lines 137, 145, 171, etc.
-Figures should be in the same order they are in the text.
-Please be more specific with words such as “quickly” “rapidly” “higher magnifications” “much larger” “quick embryogenesis” “rapid process”. If you write those words you need to provide more details how quick? How large?... etc.
-Try to describe the results for the two species in the same order. Sometimes you start with I. sinuosa and then you talk about I. rigida, sometimes you do the opposite. Being consistent helps the reader to follow the text.
-Second paragraph in this section is very hard to follow, please revise accordingly.
-Page 8 (lines 163-183) is very repetitive. Larvae information is presented twice.
-Lines 185-191 should be in the methods section.
-Lines 195-201: Include the units in your means
-Line 204: polyp instead of “polypal”
-Fecundity results could be summarized and written more clearly.
-Fig 5-6 are not referenced in the text. You need to reference every picture in your text separately. Also include the scale bars in all your photos.
-Figs 7-9 don’t look good, remove the line outside the chart, increase the font size of the legends, remove the horizontal lines.
-Fig. 1 legend: revise “Isophyllastrea”? and be consistent with the order of figures. “a” should be on the left of “b” not above. a/b and c/d are different from e/f
-Fig. 2 Image too dark, improve clarity

Discussion
-The first two paragraphs should be part of your introduction.
-Lines 228-230 are very hard to understand and is one of the most important outcomes of your research.
-Lines 231-244 are extremely hard to follow and there no connections between ideas.
-I think you should try to find environmental data (temperature/light) from La Parguera to try to relate gametogenesis to the environment.
-Why did you find such a huge difference between 2001 and 2012 for I. rigida? You don’t discuss this at all.
-You did not find planulae in 2012 for I. rigida. Any ideas?
-Lines 254-259 are not well connected.
-What do you mean by dual spawning? Partial spawning? Be more specific.
-When do eggs get infected by the symbionts?
-Lines 277-287: Your conclusion statement needs to be strengthened. What do you mean by “only mechanism capable of safeguarding their future” why?
-Are larvae produced by self-fertilization? There is no discussion on this at all.

Experimental design

-No clear how many samples were used, how statistics were performed or what the sample size is for each of the analyses performed.
-Figures need to be revised significantly.
-SEE MY COMMENTS ON BASIC REPORTING

Validity of the findings

-Information on the environment at La Parguera will strengthen this manuscript significantly. Is gametogenesis correlated with temperature and/or light intensity?
-Introduction and discussion are disorganized and hard to follow.
-Are larvae produced by self-fertilization?
-SEE MY COMMENTS ON BASIC REPORTING

Additional comments

I think this is a good study that needs a lot of work.
Coral reproduction data is hard to get and this is a very comprehensive study for two species in the Caribbean.

---

## Round 0.2 · Minor Revisions

· Academic Editor

Minor Revisions

Dear Derek and Ernesto,

Thank you for your revision, which has now been re-reviewed by one reviewer and myself. This is a very interesting study providing a very valuable new description of the sexual reproduction in Isophyllia, and I really like Table 1 and associated discussion, which provide a really good summary of sexual patterns and modes across Mussidae. This will make for a really good contribution to the field of Coral Reef Biology.

I agree with all comments from the reviewer, and will expect you to properly address them. In addition, I have provided quite a few comments and edits in the attached pdf - feel free to accept or reject appropriately. I found quite a lot of confusion with mistakes in figures referencing, wrong figure 4 uploaded (it is identical to Fig. 5), and reference format, etc, so please be vigilant in your revision.

Finally, since this study represents the first description of the sexual characteristics of I. rigida, I strongly suggest you add a new paragraph describing how I. sinuosa and I. rigida differ (or not) from each other in their gametogenesis, oogenesis, spermatogenesis, and embryogenesis stages/patterns (essentially addressing the differences and similarities between Figs. 5 and 7).

Following these corrections, I will be pleased to accept this manuscript for publication in PeerJ.

With kind regards,
Xavier

Reviewer 1 ·

Basic reporting

The authors have done a decent job addressing my earlier concerns and it is mostly ready for publication. The basic message is correct, and the content is appropriate for PeerJ. However, I would urge the authors to take another critical look at the text and edit again for clarity and brevity. To my eye, there are lots of words that can be cut without affecting meaning. For example, in the abstract the second line could start "Both species are ..". "Significantly" can br dropped throughout the abstract. Use proper sentences instead of falling in the trap of using semi-colons and adding more words. Similar issues continue to be scattered throughout the manuscript and reception of this material will be enhanced with attention to these issues. I hate reviewers writing papers the way they would want it written, and hence I have not edited for this issue. Read it through again and make up your own mind.

As I described in my first review, it is nice to see basic biology getting done, and classic tools be revisited to make new advances. I would encourage the authors to keep doing more of the same!

It would be worth revisiting the thread from Szmant (1986) regarding the association of habitat stability with reproductive mode -- I think more recent evidence indicates this trend is not well supported.

Short PLD of brooded larvae is unlikely to affect the response to high temperatures as temperates remain high whether the larvae are pelagic or they recruit as benthic polyps. This logic (e.g. Line 320) seems misleading.

Experimental design

As this is not an experimental paper, there is little need for great detail in terms of experimental design. However, critical details are still missing, for example, the depth at which the corals were sampled, and the means by which colonies were targeted for sampling, etc. It would be helpful to know seawater temperature when the corals were collected. "Sampling locations were selected for specimen abundance" is rather vague.

Results section still contains some inferences that need to hop over to the discussion. For example, the interpretation of vertical transmission (line 158), inferring seasonal synchrony (line 179), etc.

Validity of the findings

This is a nice study that reports interesting findings in the realm of basic biology. We need more of this kind of work!

---

## Round 0.3 · accepted · Accept

· Academic Editor

Accept

Dear Derek and Ernesto,

Thank you vers much for your revised manuscript which I am glad to accept for publication. I have found a couple of small edits which I will communicate to PeerJ staff. Other than that, the papier looks really good!

With kind regards,
Xavier